# Horizontal Gene Transfers in Plants

**DOI:** 10.3390/life11080857

**Published:** 2021-08-21

**Authors:** Emilie Aubin, Moaine El Baidouri, Olivier Panaud

**Affiliations:** Laboratoire Génome et Développement des Plantes, UMR 5096 UPVD/CNRS, Université de Perpignan Via Domitia, 52, Avenue, Paul Alduy, CEDEX, 66860 Perpignan, France; emilie.aubin@univ-perp.fr

**Keywords:** horizontal transfer, transposable elements, plants, genome

## Abstract

In plants, as in all eukaryotes, the vertical transmission of genetic information through reproduction ensures the maintenance of the integrity of species. However, many reports over the past few years have clearly shown that horizontal gene transfers, referred to as HGTs (the interspecific transmission of genetic information across reproductive barriers) are very common in nature and concern all living organisms including plants. The advent of next-generation sequencing technologies (NGS) has opened new perspectives for the study of HGTs through comparative genomic approaches. In this review, we provide an up-to-date view of our current knowledge of HGTs in plants.

## 1. Introduction

In living organisms, the vertical transmission of genetic information through reproduction ensures the maintenance of the self within populations and guarantees the integrity of species. Meanwhile, in most eukaryotic species, sexual reproduction also contributes to sustain adaptive diversity through the maintenance of diversity within populations. This duality of the living, i.e., the diversity of the self, is at the origin of the neo-Darwinian theory of evolution and has nourished generations of scholars in the field of evolutionary biology for over a century. In this context, horizontal gene transfers (HGTs), i.e., the transmission of genetic material across reproductive barriers, have long been considered as rare, anecdotal phenomena that could in no way constitute a major evolutionary force in multicellular organisms. However, HGTs were discovered decades ago in prokaryotes [1] and have since been shown to be major players in their adaptive process, such as in the case of the dissemination of antibiotic resistance [2]. HGTs occur at such an extent among bacteria and archaea that they impede robust phylogenetic reconstructions, leading to questioning about the taxonomic concept of species in these kingdoms [3]. In eukaryotes, unlike in prokaryotes, HGTs have long been considered as anecdotal, although the first documented cases of transfer involving a eukaryotic species were published more than 30 years ago [4,5]. Since these pioneering studies, however, hundreds of reports have evidenced the occurrence of HGTs involving protists [6] and multicellular organisms such as plants [7], animals [8], and fungi [9], therefore suggesting that, unlike it was previously thought, gene flows among distinct taxa occur frequently within ecosystems. Among all genomic components, transposable elements (TEs) appear to be particularly prone to horizontal transfers [10]. Whether this could be explained by the fact that they exist as extrachromosomal forms during the transposition cycle [11] remains to be demonstrated. Nevertheless, horizontal transfers may play an important role in the survival and dissemination of TEs in plant genomes, as we will discuss in this review.

In order to ascertain that HGTs are an important process in eukaryotes evolution, one should address both the questions of the causes (mechanisms) and the consequences (biological impact) of HGTs. Parasitism is often brought forward as an ecological niche that could favor HGTs [12]. Indeed, several documented cases of transfers concern host/parasite interactions, either directly or through an intermediate host [13], but many reports of HGTs do not involve parasitism, which leads to questioning about the alternative routes for interspecific gene flows. As for the consequences of HGTs, the main questions are whether the transferred genetic material remains functional in new genetic backgrounds and/or whether it could be at the origin of new functions, thus contributing to biological novelty. There are numerous examples of HGT-mediated acquisition of new functions in higher eukaryotes that involve plants as donors or receptors [14,15].

The latest developments in sequencing technologies (referred to as NGS: next-generation sequencing technologies) have opened new perspectives to study HGTs through genomic strategies, which lead to an acceleration of their discoveries through comparative genomic approaches (see Figure 1 for the principles of HGT detection based on sequence information). In this review, we will present the current knowledge of HGTs in plants, based on these latest discoveries. We will focus on possible mechanisms for these transfers, some of which are plant-specific, and discuss the impact of these interspecific gene flows in plant evolution.

## 2. Parasitism

As mentioned in the introduction, many cases of HGTs in eukaryotes involve host/parasite interactions, which suggests that the biological promiscuity is either necessary for or favors the transfers. Plants, as with any other eukaryotes, host a large variety of parasites from all kingdoms. Interestingly, HGTs have been evidenced between plants and viruses, bacteria, fungi, and even parasitic plants (Figure 2). Below is a brief description of some examples of such transfers.

### 2.1. Parasitic Plants

The intimate association between parasitic plants and their hosts constitutes a possible route for HGTs. In fact, parasitic plants form vascular connections with the host plant through a haustorium that enables regular transfer of water, nutrients, proteins, mRNAs, and pathogens [16]. Mower et al. [17] were the first to report a case of HGT (the mitochondrial *atp1* gene) between the parasitic genera *Cuscuta* and *Bartsia* and several *Plantago* species. Since then, several studies have confirmed that HGTs of mitochondrial genes between parasitic plants and their hosts are frequent [18,19,20]. This propensity of plants to exchange mitochondrial DNA through horizontal transfers has raised the question of the permeability of mitochondrial membranes to nucleic acids. Indeed, Koulintchenko et al. [21] found a transmembrane potential-dependent mechanism of DNA uptake into plant mitochondria. This process likely involves a voltage-dependent anion channel [22]. Whether this is the sole mechanism facilitating the movement of nucleic acids in and out of this organelle remains to be demonstrated. Transcriptomic surveys of parasitic plants have shown that transcripts could move bidirectionally between them and their hosts [23,24,25]. This suggests an RNA-based transfer mechanism where RNA is reverse transcribed into DNA before being integrated into the host genome. This should result in the absence of both introns and the promoter region of the newly integrated gene, which raises the question of the fate of such genes in the recipient species. An example of this is the transfer of a gene of unknown function from the parasitic plant *Striga to Sorghum*: Yoshida et al. [26] demonstrated that a gene encoding a 448 amino acids protein was transferred between these two species. This hypothesis (i.e., that the HGT mechanism is transcription-dependent) was, however, not validated in a recent study by Yang et al. [27] who showed that HGTs found in the genome of the parasitic species Cuscuta originated from the movement of genomic DNA. More recently, based on a comparative genomics survey of five parasitic plants with their host, Kado and Innan [28] estimated that 0.1–0.2% of the genes of obligate parasitic plants originated from HGTs from their host. Moreover, the authors showed that large genomic regions (more than 100 kbp) were transferred at once, which does not support the hypothesis of a transcription-dependent mechanism.

### 2.2. Fungi

The heterotrophic nature of fungi makes them inherently promiscuous to other eukaryotes such as plant and animals, either through symbiosis or parasitism. They may therefore be regarded as particularly prone to HGTs, which is confirmed by several reports on this matter in the literature. Interestingly, these examples show that the transfers between plants and fungi can occur in both directions. The first example of fungi-to-plant transfer is that of the group 1 intron of the mitochondrial gene *Cox1* [29,30]. Richards et al. [31] evidenced five cases of HGTs from fungi to plants using comparative genomics approaches. More recently, Wang et al. [15] evidenced that *Fusarium* head blight resistance gene *Fhb7* in wheat originated from an HGT between the *Epichloe* fungus and *Thinopyrum elongatum*, a wild relative of wheat used in wide-hybridization breeding programs to transfer the resistance. Examples of plant-to-fungi transfers are that of the *Subtilisin* gene to the pathogenic *Colletotrichum* lineage [32] and of several plant genes, including a leucine-rich repeat protein gene, known to confer pathogen resistance, to *Pyrenophora* [33]. In addition, several reports have shown that HGTs occur at a large extent among fungal lineages, thus spreading pathogenicity-related genes such as cell wall degrading enzymes [34,35]. Whether plants are involved in such important evolutionary mechanisms remains unclear.

### 2.3. Bacteria

*Agrobacterium tumefaciens* and *A. rhizogenes* are two well known plant pathogens that form root tumors upon infection through conjugative T-DNA harbored by a large tumor-inducing plasmid (Ti) or a root-inducing plasmid (Ri). In this regard, the mechanism of pathogenicity of *A. tumefaciens* and *A. rhizogenes* is HGT-dependent sensus stricto. However, as in the case of viruses, some bacterial genes can integrate into the genome of the host and, subsequently, be transgenerationally inherited. This was initially reported in *Nicotiana glauca,* which carries in its own nuclear genome a region homologous to the Ri plasmid of Agrobacterium [36]. *Linaria vulgaris* also contains sequences homologous to the T-DNA of *A. rhizogenes* corresponding to several genes, including *mikimopine synthase* (*mis*) gene and an intact and potentially functional *rolC* gene [37,38]. In sweet potato, *Agrocinopine synthase* (*Acs*), *protein C* (*C-prot*), *IaaH*, *IaaM*, *RolB*, and *ORF18* were also naturally transferred from *Agrobacterium* [39]. There is increasing evidence that natural plasmid transfer from certain *Agrobacterium* species is widespread in several plant genera other than those listed above [40]. In the near future, the analysis of hundreds of sequenced plant genomes will reveal the extent of horizontal transfer between Agrobacterium species and plant nuclear DNA.

There are other cases of more ancient transfers for which no obvious host parasite relationships could be established. Yang et al. [41] showed that the TAL-type *Transaldolase* gene from land plants originated from actinobacteria. These genes are under positive selection in several plant species and have acquired several introns following their transfer. In the marine pennate diatoms *Pseudo-nitzschia australis*, *P. granii* and *P. multiseries*, the *Ferritin* genes, used for iron storage, are more closely related to that of archaebacteria than other plants, suggesting their replacement in this lineage through HGT [42]. Finally, Metcalf et al. [43] showed the multiple transfer of the antibacterial gene *Glycosyl hydrolase 25 muramidase* from bacteria to plants (but also fungi, animals and archaebacteria), thus evidencing the widespread dissemination of a gene originally involved in the survival of the bacteria in competitive growth throughout the tree of life.

### 2.4. Viruses

Viruses are well known plant pathogens that are good candidates to search for HGTs. There are some examples in the literature suggesting that transfers from viruses to plants may exist. Chen et al. [44] evidenced the presence of endogenous pararetroviral-like sequences in the genome of rice *O. sativa*. These sequences are homologous to tungro virus, a rice pathogen that causes significant yield losses worldwide. A similar observation was made for the florendoviruses, the infection of which has left several viral-like sequences in the genome of several plant species [45]. The fate of these transferred sequences, whether they are at the origin of new functions, as was recently evidenced in mammalian genomes [46], or only remain in their recipient genome as fossils, has to be further investigated. In one instance, however, two viral genes, i.e., encoding the capsid protein and the RNA-dependent RNA polymerase, were horizontally transferred to various eukaryotic genomes (including that of the two plant species *Arabidopsis thaliana* and *Festuca pratensis)* and remained functional in several lineages, suggesting their possible exaptation by their hosts [47].

**Figure 2 life-11-00857-f002:**
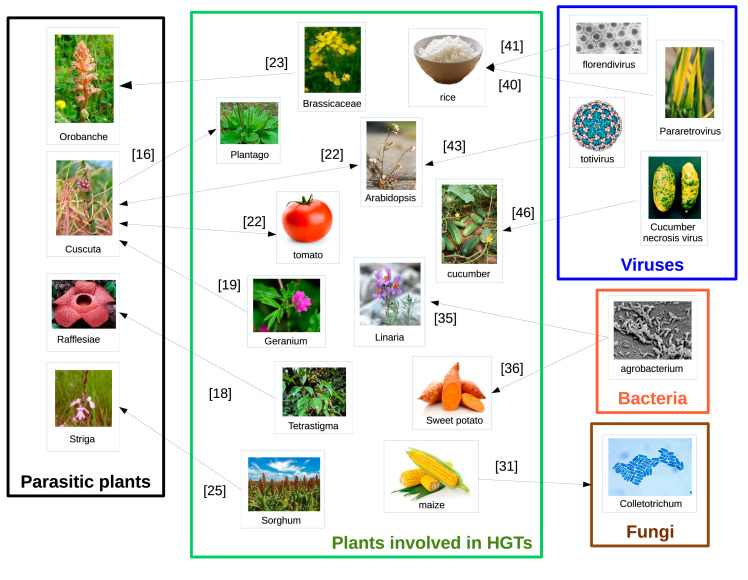
Some examples of HGTs in plants involving parasitism. Numbers between brackets correspond to the references cited in the manuscript.

The transfer from a host to its parasite is only the first stage of a long journey until it reaches a recipient species where it could eventually contribute to increase its adaptiveness. Viruses are, therefore, also of interest as potential vectors for plant-to-plant HGTs. Ghoshal et al. [48] showed that the cucumber necrosis virus could encapsidate TEs (LTR-retrotransposons and LINEs) from its host *Nicotiana benthamiana* upon infection, although the genetic material was in the form of RNA and no direct evidence of a transfer to another species was provided. Gilbert et al. [13] provided stronger evidence for the possible role of viruses as HGT vectors. By conducting deep sequencing of 21 moth baculovirus populations, the authors showed that 4.8% of the viruses contained at least one host sequence.

## 3. Grafting

Grafting, consisting of joining the vascular tissues of two different plants, has been known as a common practice in horticulture for centuries and has been exploited economically worldwide for many species. Experimental grafting was developed over a century ago as a tool to study the movement of molecules within the plant [49]. This technique allowed researchers to demonstrate that not only hormones or metabolite could circulate through vascular tissues but also genetic information in the form of mRNAs [50], small RNAs [51], and even genomic DNA [52,53]. In a pioneering study, Stegemann and Bock [54] conducted grafting experiments between two *Nicotiana tabacum* mutants, carrying both an antibiotic resistance and a reporter gene, each being distinct between the two mutants. Using this elegant screen, the authors evidenced that some cells that they had isolated from the graft site exhibited a resistance to both antibiotics together with a double fluorescence, indicative of a fusion of genetic material between the two mutants upon grafting. Using a similar screen, Stegemann et al. [52] further showed that the complete transfer of a chloroplast through grafting was possible between two distinct species of *Nicotiana* (i.e., *N. glauca*/*N. tabacum* and *N. benthamiana*/*N. tabacum* grafts). In this case, the transferred plastid replaced that of the recipient species, a mechanism referred to as plastid capture. In a recent report from the same group, Fuentes et al. [53] showed that allopolyploid plants could be regenerated from *calli* derived from the graft site in a grafting experiment between *Nicotiana glauca* (a tree species) and *Nicotiana tabacum* (a herbaceous species). Interestingly, these plants exhibit a stable karyotype, can harbor fertile flowers, and produce viable seeds. Therefore, the authors demonstrated that grafting could be at the origin of allopolyploid species. Since natural grafting has also been described in natural ecosystems [55], this strongly suggests that it could be a mechanism of HGT in plants [56].

## 4. Food Chain

The hypothesis, referred to as “you are what you eat”, proposed by Doolittle [57], posits that HGTs could also occur naturally along the food chain between prokaryotes and phagotrophic eukaryotes, the latter feeding on the former. Evidence supporting this hypothesis is scarce: probably the most spectacular example of such transfer is that of the sea slug *Elysia chlorotica*. This animal becomes photosynthetic when feeding on the algae *Vaucheria litorea* through the acquisition of its chloroplasts [58]. This transfer from a plant to an animal is, however, not transgenerational because these chloroplasts are not transmitted to the egg of the slug [59] and can therefore not be considered as a true HGT, but it shows that genetic material can be transmitted throughout the food chain and remain functional.

## 5. Other Routes

There are many cases where HGTs have been evidenced without any obvious explanation regarding their mechanism. In a vast majority, as in the case of the documented HGTs cited above, these concern organellar DNA and, in particular, mitochondrial genes, such as *cox1* [60,61], *nad1* [62,63], or the ribosomal protein genes *rps2* and *rps11* [64]. The most striking case of mitochondrial DNA transfer concerns the entire genome of several algae and a moss mitochondria into that of the angiosperm *Amborella* leading to a total size of 3.9 Mbp, i.e., six genome equivalent of their average size in flowering plants [65,66]. The mechanism of such large transfers could be through the fusion of the whole mitochondria [66].

Evidence of HGTs of nuclear genes involving organisms with no obvious biological relationships is even more scarce. Whether this is due to detection difficulties or because they are indeed rare events remains to be clarified. There are, however, several examples of well-established cases of such transfers in a comprehensive comparative study of 434 transcriptomes and 40 genomes. Li et al. [67] showed that the adaptation of ferns to low light condition (i.e., under the canopy of angiosperms) was enabled by the HGT of a neochrome gene from bryophytes. This transfer was dated at 179 Mya, which is significantly more recent than the split of the two lineages (>400 Mya). Similarly, Christin et al. [68] showed that the *PPC* gene (the C4 carbon-fixing enzyme phosphoenolpyruvate carboxylase) has been transferred several times in plants, giving rise to independent acquisition of C4 photosynthetic metabolism. This was confirmed by a comparative genomics survey between the grass species *Alloteropsis semialata* and 146 species from the same taxon that evidenced the occurrence of a total of 59 HGTs involving at least nine donor species [69]. Interestingly, these transfers are clustered into large genomic blocks of up to 170kb, similar to what was reported by Kado and Innan [28] for parasitic plants. The genome analysis of the most ancestral land plant, the moss *Physcomitrella patens* has provided evidence that multiple HGTs have enabled the acquisition of new functions associated to land colonization, such as xylem formation, plant defense, and nitrogen recycling as well as the biosynthesis of starch, polyamines, hormones, and glutathione [70]. These last examples show that HGTs within the green lineage may play an important role in plant evolution, thereby creating adaptive biological novelty. However, plants may have also benefited from HGTs involving more distantly related taxonomic groups such as animals. This is the case of the transfer of two *Transferrin* genes from insects to *Theobroma cacao* that may have contributed to the acquisition of new functions related to iron homeostasis, immunity, cell growth, and differentiation in this important crop species [71]. Transfers from plants to animals also appear to be associated with the emergence of new functions: Drosomycin-type antifungal peptides (DTAFPs) are widespread in plants. The corresponding genes have also been found in several animals, although their distribution in this kingdom is patchy. Zhu and Gao [72] showed that the *DTAFP* genes in animals originated from plants through HGT. Hespeels et al. [73] showed that four *trehalose-6-phosphate synthase* (*TPS*) genes, known to be involved in desiccation resistance in rotifers, are of plant origin. More recently, Xia et al. [14] showed that the whitefly *Bemisia tabaci* has acquired horizontally the plant-derived phenolic *glucoside malonyltransferase* gene *BtPMaT1*, which allows this insect to neutralize phenolic glucosides (a toxin synthesized by plants in response to insect feeding).

## 6. Horizontal Transfers of Transposable Elements (HTTs)

Plant genomes, in the same way as higher eukaryotes, are mainly composed of transposable elements, referred to hereafter as TEs [74]. TEs are of two main classes: class I elements, the retrotransposons, transpose via an RNA intermediate, while class II elements, the transposons, transpose via a DNA intermediate [75]. TEs from both classes have in common that they can be found as extrachromosomal forms in the cell of their host at one point of their transposition cycle [11]. This suggests that these genomic components may be more prone to horizontal transfers than the genes per se [76]. In addition, most TEs are inactive in plants because transposition is strictly controlled by several silencing pathways at both transcriptional and post-transcriptional levels [77]. Moreover, several studies have shown that TE-related sequences are quickly eliminated from their host genome through deletions and recombinations [78]. The combined action of silencing and elimination should therefore lead to the complete elimination of TEs from most species, which is exactly the opposite of what is observed. In this context, horizontal transfers of transposable elements (HTTs) could be a mechanism allowing the survival of TEs, as an escape from the silencing and elimination in their host genome and a transfer to a “naïve” genome where they could propagate before being, in turn, silenced [79]. The first discovery of HTT in eukaryotes was that of the P element in *Drosophila* [80]. Hundreds of cases of HTTs have since been reported in both prokaryotes and eukaryotes, which suggests that TEs may indeed be more prone to horizontal transfers. However, the wide diversity of TE types in eukaryotic genomes [75], the fact that they do not belong to the gene space of their host genome and, therefore, diverge at a higher rate than genic sequences, and their propensity to multiply their copy number while active, especially in plants, make HTT detection inherently difficult [81]. Out of the three detection criteria described in Figure 1, HS (high similarity) is the most commonly used when HTTs are searched from genomic data such as DNAseq or RNAseq. This is the case for the detection of an HT of a MULE transposon between rice and millet [82], of the LTR-retrotransposon *Route66* among several grass genomes [83], of the LTR-retrotransposon *Copia25* across angiosperms [84], of the tomato retrotransposon *Rider* between *Brassicaceae* and *Solanaceae* [85], of a *Penelope-Like* retroelement from an arthropod to conifers [86], and of the non-LTR *AdLINE3* from an arthropod to a peanut [87]. In these examples, HTTs were evidenced from homology searches using a particular TE as a query on nucleotide databases. In addition, when a TE family is widespread in taxonomic groups, the PI criteria can also be applied, as in the case of the HTT of *PIF*-like transposons in *Triticeae* [88], the *Mothra* helitron in angiosperms [89], or the centromeric retrotransposons in grasses [90]. With the advent of NGS-based plant genome sequencing projects over the last decade, one could tentatively search for HTTs based on whole genome similarity searches. This, however, remains too computationally intensive and raises some conceptual issues, such as the false positive detection of house-keeping genes, for which the high sequence identity among distantly related taxa results from a strong purifying selection through a strict vertical inheritance, rather than horizontal transfers. However, hundreds (if not thousands) of TE families have been characterized from these genomic resources, making possible the search for HTTs specifically. El Baidouri et al. [91] followed such a strategy by first mining out LTR-retrotransposons from 42 sequenced and assembled plant genomes, defining families based on a two-step homology clustering and, finally, detecting families containing elements from distantly related species. The authors thus evidenced the occurrence of 32 HTTs among this sample of 40 angiosperm species, which led them to estimate that hundreds of thousands of HTTs had occurred among flowering plants within the last two million years, therefore providing strong evidence that HTTs are important for the survival of TEs in plant genomes and suggesting that interspecies gene flows in ecosystems are frequent.

## 7. Conclusions

It is now widely admitted that HGTs are widespread in eukaryotes and this review provides some examples of HGTs where plants are involved. As mentioned in the introduction, the open questions regarding HGTs concern mainly the mechanisms that enable gene flows across distinct taxonomic groups and their biological impact. As for the mechanisms, it is clear that the majority of HGTs concern species that live in biological promiscuity. In this regard, it is not surprising to observe HGTs involving viruses, bacteria, or fungi, while plant-to-plant transfers evidenced so far concern either parasitic plants or grafting. The exchange of genetic material through physical contact therefore appears as the main mechanism. However, the next step towards our understanding of interspecific gene flows is to unravel how HGTs could occur between species with no biological relationships and, in particular, whether parasites may act as vectors, or “genetic bridges”, to spread adaptive genes among sympatrical species within or across kingdoms, thereby contributing to the adaptation of ecosystems to environmental changes.

HTTs are a particular type of HGTs. TEs may indeed be more prone to HGTs because they can be found as an extrachromosomal form during their transposition cycle. Moreover, transposition is in some cases triggered by biotic stress in plants [92]. One could, therefore, anticipate that pathogen attacks may favor HTTs, although this remains to be tested. As for their biological impact, several of the reports that we discussed clearly showed that HGTs are associated with new functions, as in the case of the transfer of an antimicrobial gene [43] or of the detoxification gene from plants to insects [14]. These are often ancient inter-kingdom HGTs that are relatively more easy to detect than recent ones. The next challenge is, therefore, to develop new methods for the detection of recent biologically relevant HGTs. These methods will have to rely on the use of a combination of omics approaches, whereby the detection of HGTs could be achieved through similarity searches at full genome scale in large samples, while transcriptomic data could provide clues on the biological fate of the transferred genes. Finally, the constant development of new sequencing technologies, resulting in lower operating costs and higher throughput, opens new perspectives in the in situ study of gene flows in ecosystems, where population genomics approaches could allow researchers to quantify their extent in natura and their potential role in adaptation.

## Figures and Tables

**Figure 1 life-11-00857-f001:**
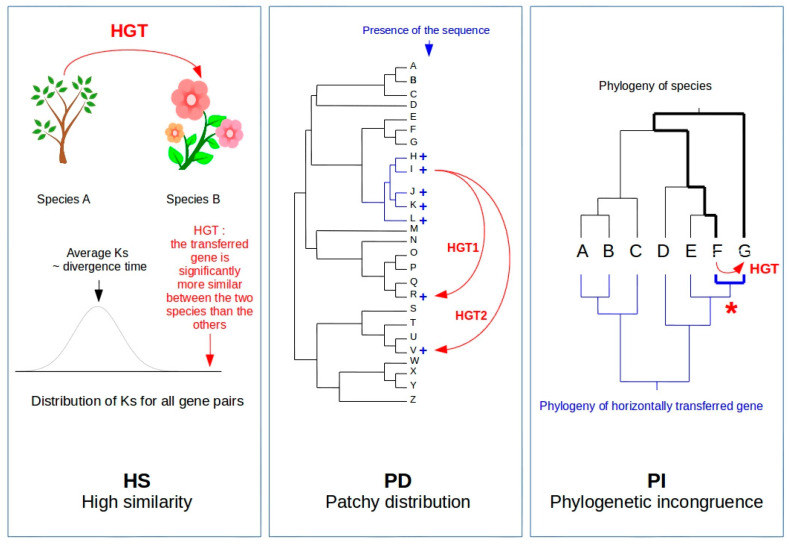
The three criteria used for HGT detection through comparative genomics: high similarity (HS), phylogenetic incongruence (PI) and patchy distribution in phylogenies (PD) [9]. The HS criterion requires access to genomic data of the two species involved in the transfer. It consists of establishing that gene homologs between two species exhibit a sequence identity that is significantly higher than the average of the other homologs in the genome, therefore not originating from vertical transmission. In order to circumvent the possible effect of strong selection (that could lead to high sequence identity), synonymous substitutions are used to measure sequence divergence. The PI criterion is based on the incongruence between the topologies of the phylogenetic tree of the species and that of the horizontally transferred genes. This requires sequence information for a large phylogenetic panel, as well as to identify the presence of the transferred gene in a large enough species sample. The PD criterion is based on the presence of a given sequence in only a subset of species across a phylogenetic tree. The presence of a sequence thus shared by phylogenetically distant species, albeit not by more closely related taxa, could suggest the occurrence of an HGT. This third criterion must, however, be taken with caution since patchy distribution in phylogenies may also be caused by gene losses. Both PI and PD criteria require access to genomic information on a phylogenetically relevant sample of taxa, meaning that plant material should be available and subsequent molecular analyses completed, which is not always the case. On the contrary, the availability of large genomic public datasets, made possible by the development of NGS, opens new opportunities for the detection of HGT through bioinformatic methods. These strategies mostly use the HS criterion because phylogenetic trees are far from being saturated with genomic data, therefore excluding the systematic use of both PI and PD criteria. Figure legend: Curved red arrows represent cases of HGTs. HS: Ks = synonymous substitution rate. PD: phylogenetic tree of 26 species + = presence of the sequence involved in the HGT. PI: = node showing the phylogenetic incongruence because of the close relatedness of the gene transferred between taxa F and G. Bold lines illustrate the difference in the phylogenetic distance between taxa F and G and the nucleotidic distance between the horizontally transferred genes.

## Data Availability

Not applicable. No data was generated for this study.

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
