# Peer review of "Horizontal Gene Transfers in Plants"

_life, 2021, doi:10.3390/life11080857_

Round 1

Reviewer 1 Report

Comments to Authors.

The manuscript submitted by Aubin et al. is a short but updated review of HGT events from other organisms to plants, from plants to other organisms and among plants that deserve to be published in a Plants special issue of Life.

 I have only some comments in the aim to improve the manuscript.

  1. In relation to the discussion of horizontal gene transfer between parasitic plants, an introduction to the possible role of haustoria in the transfer mechanism (Sanchez.Puerta el al New Phytol. 214 (2017) 376-387) could be welcome.
  2. In figure 1, arrow shows the transfer from fungi to plants but in this case, direction of transfer is from plant to fungi as it said in the main text.
  3. In relation to acquisition of genes by phagotrophy, authors should remark that in this case, direction of transfer will be ever from plants to animals.
  4. Discussing the Amborella case, it should be interesting to discuss the possible mechanism by mitochondria fusion suggested by Rice in ref. 62.
  5. Manuscript fails in discuss some case involving plastidic genes.

Other minor comments:

  • In line 53, I suggest to change “… that involve plants” to “ …. that involves plants as donors or receptors”
  • In line 990-91, I suggest to change “HGT have been evidenced in……..parasitic plants” to “HGT have been evidenced between plants and ……… and other parasitic plants”.
  • I suggest to change the caption of figure 1 to “Some examples of HGT in plants involving parasitism”
  • Reference 46 in no cited in the main text. I suppose that correspond to second 45 in the page 5
  • Please check the last references. Reference 88 is no cited in the main text, but it appears pated to ref 83 in the reference list.

Author Response

  1. In relation to the discussion of horizontal gene transfer between parasitic plants, an introduction to the possible role of haustoria in the transfer mechanism (Sanchez.Puerta el al New Phytol. 214 (2017) 376-387) could be welcome.

We have added a sentence regarding this matter as an introduction to the paragraph.

  1. In figure 1, arrow shows the transfer from fungi to plants but in this case, direction of transfer is from plant to fungi as it said in the main text.

Correction has been made.

  1. In relation to acquisition of genes by phagotrophy, authors should remark that in this case, direction of transfer will be ever from plants to animals.

We have added a sentence (l226)

  1. Discussing the Amborella case, it should be interesting to discuss the possible mechanism by mitochondria fusion suggested by Rice in ref. 62.

we added a sentence (l238).

  1. Manuscript fails in discuss some case involving plastidic genes.

We haven't discussed HGTs invoving plastidic genes because, to our knowledge, these are essentially intra-cellular transfers.

Other minor comments:

  • In line 53, I suggest to change “… that involve plants” to “ …. that involves plants as donors or receptors”

We have made the correction. 

  • In line 990-91, I suggest to change “HGT have been evidenced in……..parasitic plants” to “HGT have been evidenced between plants and ……… and other parasitic plants”.

We have made the correction.

  • I suggest to change the caption of figure 1 to “Some examples of HGT in plants involving parasitism”

This has been vhanged accordingly

  • Reference 46 in no cited in the main text. I suppose that correspond to second 45 in the page 5

thanks for noticing this. It has now been corrected.

  • Please check the last references. Reference 88 is no cited in the main text, but it appears pated to ref 83 in the reference list.

reference 88 is cited in line 334

Reviewer 2 Report

This study made a review of the recent important information on the mechanisms, and biological roles of horizontal gene transfer in plants. The topic is very interesting and it is important to understand the consequences of HGT and summarize current knowledge from this area of research. The manuscript is clear, unambiguous, and well-written and it logically reviews and discusses important recent literature in this field. The drawings sum up the research in a very interesting form.

However, there are several problems that deduct from the quality of this manuscript. Below are several comments on this work.

  1. In my opinion, Authors should avoid certain informal expressions, e.g.

line 19 - „the self”

line 22 – „diverity of the self”

  1. Write all genes / alleles in italics, but not genes products, e.g.

line 132 – subtilisin

line 144 – mikopin syntase

lines 145, 148, 151 etc.

  1. Is writing a fragment lines 112-120 certainly should be in italics? See also line 237.
  2. Line 130 – Epichloe is a genus – should be write in italics.

Taking the above into consideration, I recommend this manuscript for publication with a minor revisions.

Author Response

  1. In my opinion, Authors should avoid certain informal expressions, e.g.

line 19 - „the self”

line 22 – „diverity of the self”

We use the term "the self" in reference to the phrasing of C. Darwin in "the origin of species". Here, the self is referred to as all members of the same species, i.e. that can cross and produce fertile progenies.

  1. Write all genes / alleles in italics, but not genes products, e.g.

We have now corrected all these typos.

  1. Is writing a fragment lines 112-120 certainly should be in italics? See also line 237.

This was corrected.

  1. Line 130 – Epichloe is a genus – should be write in italics.

THis has been corrected.

Reviewer 3 Report

The presented review is devoted to the current topic of horizontal gene transfer. It is well structured, well illustrated, and easy to read. However, some points are not written accurately. First of all, this concerns the section devoted to genes transferred from Agrobacterium. In this section, the name of opine is not spelled correctly. There should be mikimopine. It is not correct to say that the gene transfer was from A. tumefaciens, since most genes in naturally transgenic plants are homologous to those from A. rhizogenes. References to the primary sources are also incorrect. For the first time, homologues of T-DNA genes in plants were described by White et al (https://doi.org/10.1038/301348a0)T-DNA in Linaria was first described by Matveeva et al. (https://doi.org/10.1094/MPMI-07-12-0169-R) It contains intact rolC, while mis there is mutant. And today there are dozens of species of natural GMOs.(https://orcid.org/0000-0001-8569-6665 https://doi.org/10.1007/s11103-019-00913-y)
A subtle point concerns retrotransposons. If the infectious nature is described, then it is correct to talk about retroviruses. The difference between viruses and transposons is rather arbitrary. This should be borne in mind by assigning the appropriate sections.

Author Response

The presented review is devoted to the current topic of horizontal gene transfer. It is well structured, well illustrated, and easy to read. However, some points are not written accurately. First of all, this concerns the section devoted to genes transferred from Agrobacterium. 
In this section, the name of opine is not spelled correctly. There should be mikimopine. 
->  thank you for pointing out this typo, it has been corrected in the manuscript.

It is not correct to say that the gene transfer was from A. tumefaciens, since most genes in naturally transgenic plants are homologous to those from A. rhizogenes. 
->This is absolutely true and corrections have been made in the manuscript

References to the primary sources are also incorrect. For the first time, homologues of T-DNA genes in plants were described by White et al (https://doi.org/10.1038/301348a0)T-DNA in Linaria was first described by Matveeva et al. (https://doi.org/10.1094/MPMI-07-12-0169-R) It contains intact rolC, while mis there is mutant. And today there are dozens of species of natural GMOs.(https://orcid.org/0000-0001-8569-6665 https://doi.org/10.1007/s11103-019-00913-y)
-> These references have been included in the current review version

A subtle point concerns retrotransposons. If the infectious nature is described, then it is correct to talk about retroviruses. The difference between viruses and transposons is rather arbitrary. This should be borne in mind by assigning the appropriate sections.
-> In fact, even in the case of horizontally transferred retrotransposons, there is no documented evidence of induced pathogenicity. Therefore, we feel that the difference between retroviruses and retrotransposons is clear and should be kept as such. 

Round 2

Reviewer 3 Report

The main remarks have been taken into account by the authors. The review can be accepted.